# A Baculovirus Expression Vector Derived Entirely from Non-Templated, Chemically Synthesized DNA Parts

**DOI:** 10.3390/v15101981

**Published:** 2023-09-23

**Authors:** Christopher Nguyen, Amanda Ibe-Enwo, Jeffrey Slack

**Affiliations:** 1Stylus Medicine, Inc., 200 Berkeley St., Boston, MA 02116, USA; chris.nguyen@stylusmedicine.com; 2Voyager Therapeutics, 64 Sidney St., Cambridge, MA 02139, USA; audumma@vygr.com

**Keywords:** baculovirus expression vector, synthetic biology, rAAV, Golden Gate assembly, synthetic virus genome

## Abstract

Baculovirus expression system1s are a widely used tool in recombinant protein and biologics production. To enable the possibility of genome modifications unconstrained through low-throughput and bespoke classical genome manipulation techniques, we set out to construct a baculovirus vector (>130 kb dsDNA) built from modular, chemically synthesized DNA parts. We constructed a synthetic version of *Autographa californica* multiple nucleopolyhedrovirus (*Ac*MNPV) through two steps of hierarchical Golden Gate assembly. Over 140 restriction endonuclease sites were removed to enable the discrimination of the synthetic genome from native baculovirus genomes. A head-to-head comparison of our modular, synthetic *Ac*MNPV genome with native baculovirus vectors showed no significant difference in baculovirus growth kinetics or recombinant adeno-associated virus production—suggesting that neither baculovirus replication nor very-late gene expression were compromised by our design or assembly method. With unprecedented control over the *Ac*MNPV genome at the single-nucleotide level, we hope to ambitiously explore novel *Ac*MNPV vectors streamlined for biologics production and development.

## 1. Introduction

Baculoviruses are a diverse family of insect viruses that have been extensively studied due to their original use in the biological control of insect pests and their later safe and effective application in human gene therapies and vaccines [1]. The baculovirus expression vector (BEV) system biotechnology has been used for decades in the routine production of recombinant proteins in insect cell lines in academia and industry. This paper describes a major advancement of the BEV system and extends its capabilities for producing more complex biologics such as virus-like particles or recombinant adeno-associated viruses (rAAVs).

An extensive and versatile manipulation of baculovirus genomes is needed for the development of next-generation BEVs with enhanced cell line infectivity, superior biologics production, control of production timing, or lowered contaminants. Past BEV systems have been devised for the insertion of heterologous sequences into a baculovirus genome (usually at the *polyhedrin* locus), including recombination in insect cells [2], homologous recombination in yeast [3], Cre-lox recombination [4], Tn7 transposition [5], and direct ligation into unique restriction sites within the genome [6]. Historically, baculovirus genomes have been manipulated through classical methods such as homologous recombination in insect cells or Red recombination in *E. coli* [7,8]. The fundamental limitation of these methods is their restriction to a single locus modification with each mutagenesis. Thus, the modification of multiple loci across the genome is a slow, successive process. Another fundamental limitation is the non-modular nature of this iterative, bespoke genetic modification workflow—that is, if there are multiple desirable versions of more than one locus, a long-term effort must be expended to generate this specific genome configuration.

Recently, novel approaches have emerged for the assembly of large virus genomes from modular DNA parts. These have included Herpesviridae [9,10], Baculoviridae [11], and Poxviridae [12]. These assembly approaches have relied on homologous recombination in vivo either in yeast or in permissive cells. In one instance, researchers assembled an infectious baculovirus genome in vitro from PCR-amplified fragments. 

In this study, we devised an in vitro genome assembly system to support the generation of an *Autographa californica* multiple nucleopolyhedrosis virus (*Ac*MNPV) genome entirely from chemically synthesized DNA parts. We also demonstrate the application of this synthetic BEV and its cloning technology by producing rAAVs in insect cells.

### Rationale

The assembly methods for the reconstitution of large DNA virus genomes have typically relied on in vivo homologous recombination between large DNA fragments, either in yeast or in permissive cells [9,10,11,12]. At least one instance of the in vitro construction of a baculovirus genome through Gibson Assembly [13] followed by isolation in *E. coli* has been documented [14].

Another technique for in vitro DNA assembly is Golden Gate assembly (GGA), which is based on successive rounds of cleaving by Type IIS restriction enzymes and ligation by T4 DNA ligase [15]. Commercial reagents and protocols for GGA have been developed, enabling the assembly of DNA molecules from 24 [16,17], or even as many as 45 [18], constituent fragments.

Because the simultaneous assembly of multiple DNA fragments through Golden Gate assembly (GGA) is efficient and has high fidelity, this technique enables a high degree of modularity in a synthetic genome design while reducing workload through reducing the number of hierarchical assembly steps between fundamental synthetic fragments and the final genome. We therefore chose two levels of hierarchical Golden Gate assembly as the method for the construction of a synthetic baculovirus genome.

Because of its widespread use as the backbone for the BEV used in protein expression and biologics manufacturing, we chose to base the design of a synthetic baculovirus genome on the E2 strain of *Ac*MNPV (Accession No. KM667940.1) [19].

## 2. Materials and Methods

### 2.1. Insect Cells

*Spodoptera frugiperda* cells (*Sf9*) were cultured in ESF AF™ Insect Cell Culture Medium (#99-300-01, Expression Systems, Davis, CA, USA). 

### 2.2. Virus Stocks

Baculovirus stocks were stored as baculovirus-infected insect cells (BIICs). BIIC stocks were generated by transfection of *Sf9* cells with bacmid DNA using TransIT-Insect Transfection Reagent (# MIR 6100, Mirus Bio, Madison, WI) as follows: A 30 mL culture of *Sf9* cells (2 × 10^6^ cells/mL) was prepared via the dilution of seed train into an ESF medium. Bacmid-transfection complexes were prepared by combining 200 ng of bacmid DNA, 5 µL of TransIT Reagent, and 200 µL of non-supplemented Grace’s Salts (#G8142, Sigma-Aldrich, St. Louis, MO, USA), followed by incubation at room temperature for 20 min. The entire mixture was then added directly to the *Sf9* culture to initiate the reconstitution of infectious baculovirus.

At 3 days post-transfection, the cells were pelleted via centrifugation (5 min, 300× *g*), and virus-containing supernatant was transferred to a separate tube and stored in the dark at 4 °C. 

### 2.3. Sf9 Reporter Cell Line for Detection of Baculovirus Infection

Recombinant *Sf9* cells encoding a green fluorescent protein (GFP) under the control of a baculovirus-inducible promoter were generated by a method similar in principle to that shown by Hopkins and Esposito [20]. *Sf9* cells were transfected with a plasmid harboring a GFP expression cassette, whose transcription is controlled by the baculovirus *39k* promoter, which requires transactivation by the baculovirus IE1 protein for activity [21]. In brief, 5 µg of plasmid DNA was combined with 15 µL of TransIT reagent in 200 µL of non-supplemented Grace’s Salts. Plasmid-TransIT mix was incubated for 20 min at room temperature and added to a 30 mL culture of *Sf9* cells (2 × 10^6^ cells per mL). At 48 h post-transfection, puromycin (#J67236,Alfa Aesar, Haverhill, MA, USA) was added to the culture to a final concentration of 5 µg/mL. The stable integration of the inducible GFP cassette into the *Sf9* genome was achieved through the continual passaging of the cells for three weeks in the presence of puromycin. These polyclonal reporter cells were afterwards maintained in ESF medium without puromycin. 

### 2.4. Baculovirus Titration by TCID50 Assay Using Sf9 Reporter Cell Line

Infectious units of baculovirus were quantitated using tissue culture infectious dose 50% (TCID50) assay in 96-well format. Virus samples were 10-fold serially diluted in ESF medium, and 0.1 mL of each dilution was used to infect 8 wells, each containing 6 × 10^4^
*Sf9* reporter cells (described above). After 6 days, the wells were scored as positive or negative through viewing them under fluorescence microscopy. A well was scored positive if any plaques or foci composed of GFP-expressing cells were observed. Finally, positive and negative wells were used to calculate the TCID50 value for each sample according to the Spearman–Kärber method [22,23], and final TCID50/mL values for each sample were recorded. 

### 2.5. Titration of Packaged Recombinant AAV Vector Genome Copies by Quantitative Polymerase Chain Reaction

Recombinant AAV (rAAV)-containing samples were lysed and clarified of cell debris as follows: 1 mL of whole insect cell culture containing both cells and medium were lysed by the addition of Triton X-100 to 0.5% (*v*/*v*) finally, L-arginine to 2 mM finally, and benzonase (#E1014, MilliporeSigma, Burlington, MA, USA) to 10 enzyme units/mL finally. The solution was vortexed extensively and set to rotate at 37 °C for 1 h for benzonase digestion. The mixture was then clarified of cell debris through centrifugation at 4000× *g* for 5 min, and the supernatant was transferred to a new container.

The clarified lysate was then subjected to DNase I digestion to remove any non-encapsidated DNA as follows: 5 µL of clarified lysate was combined with 30 µg of DNase I enzyme (#3D1401, Teknova, Mansfield, MA, USA) in a 100 µL final volume of 1× DNase buffer [1 mM MgCl2, 1 mM CaCl2, 1 mM Tris pH 7.5] and incubated at 37 °C for 1 h. Simultaneous DNase I-inactivation and capsid digestion were then performed through the addition of 120 µg Proteinase K enzyme (Teknova, #P2050) and the addition of EDTA (pH 8.0) to an 11 mM concentration and Proteinase K buffer to a 1× final concentration (100 mM NaCl, 0.1% sodium dodecyl sulfate) and a final volume of 225 µL. Proteinase K-based digestion of capsids was carried out at 55 °C for 1 h, followed by heat-denaturation at 95 °C for 10 min and a 4 °C hold.

DNase-I/Proteinase-K-treated samples were then diluted 1:40 in 10 mM Tris pH 7.5 (5 µL sample + 195 µL diluent). A Plasmid DNA standard was also prepared via 10-fold serial dilution in 10 mM Tris pH 7.5. Finally, 4 µL of diluted sample or plasmid standard was subjected to quantitative polymerase chain reaction (qPCR) in a 20 µL reaction containing 1× Taqman^®^ Fast Advanced Master Mix (#4444557, ThermoFisher, Waltham, MA, USA), 300 nM forward primer, 300 nM reverse primer, and 150 nM probe targeting the CMV promoter. Reactions were performed using the Fast” setting on a QuantStudio3 Real-Time PCR System (#A28567, ThermoFisher). Cycle threshold (Ct) values vs. plasmid DNA copy number inputs were plotted to obtain linear regression, enabling Ct-to-copy-number conversion. In all cases, qPCR efficiency was confirmed to be within 95 to 105%. Packaged vector genome copies (i.e., DNase-resistant CMV copies) per mL of clarified lysate were back-calculated from the CMV copy number in the qPCR reaction as follows:
$$\begin{array}{l} \mathrm{Titer}(\mathrm{packaged}\mathrm{vector}\mathrm{genomes}/\mathrm{mL}\mathrm{clarified}\mathrm{lysate})\\ \;\;\;=\mathrm{qPCR}\mathrm{copies}/4\;\mu L\times 225\;\mu L/5\;\mu L\times 200\;\mu L/5\;\mu L\times 1000\;\mu L/1\mathrm{mL} \end{array}$$

### 2.6. Bacmid DNA Preparation

Purified bacmid DNA stocks were extracted from midiprep-scale 200 mL overnight-shaken (37 °C) *E. coli* cultures using the NucleoBond Xtra Midi kit (#740410, Macherey-Nagel, Bethlehem, PA, USA), according to the manufacturer’s instructions. 

### 2.7. Polymerase Chain Reaction

All polymerase chain reactions (PCRs) were performed using Q5 2X Hot Start Mastermix (#M0494, New England Biolabs, Ipswich, MA, USA) according to the manufacturer’s instructions. Primers were used at a final concentration of 0.5 µM. 

### 2.8. Bacterial Transformation

Bacmids constructed in vitro via Golden Gate assembly were electroporated into NEB^®^ 10-beta *E. coli* (#C3020K, New England Biolabs, Ipswich, MA, USA) according to the vendor’s instructions. Plasmids modified in vitro were transformed into chemically competent NEB^®^ 5-alpha *E. coli* (#C2987H, New England Biolabs) according to the vendor’s instructions. Plasmids harboring subgenomic ‘A’ regions were electroporated into electrocompetent CopyCutter^TM^
*E. coli* cells (#EPI400, Lucigen Corporation, Middleton, WI, USA) according to the vendor’s instructions. 

### 2.9. Design of a Golden Gate Assembly-Compatible Baculovirus Genome

#### 2.9.1. In Silico Removal of Type IIS and Type II Restriction Enzyme Sites from the *Ac*MNPV Genome

The recognition sites of several Type II and Type IIS restriction enzymes were removed from the *Ac*MNPV genome at the design stage as follows: In order to minimize the risk of disrupting essential *cis*-elements for baculovirus function and replication, each enzyme site was removed by cross-referencing closely related baculovirus genome sequences in the NCBI database. A roughly 206 bp region comprising the 100 bp flanking each restriction enzyme site was queried against the non-redundant nucleotide database filtered on *Baculoviridae* (taxid: 10442). Generally, the sequence record with highest % identity with the query, yet lacking the enzyme site in question, was used as a cross-reference for how to remove the enzyme site in the *Ac*MNPV sequence (Figure 1B). Within known *Ac*MNPV open reading frames, maintaining the protein sequence was prioritized over a non-silent variation in each cross-reference. The Type IIS restriction endonuclease (REN) enzyme sites removed include BsaI, BsmBI, and PaqCI (Figure 1A). The Type II REN enzyme sites removed included AgeI, KpnI, NheI, BamHI, and several other enzyme sites.

#### 2.9.2. Decomposition of Domesticated *Ac*MNPV Genome into Subgenomic ‘A’ Regions and ‘B’ Subfragments

To decompose the domesticated *Ac*MNPV genome above into subgenomic ‘A’ regions and ‘B’ subfragments, we first chose an 18-member set of 4-base overhangs predicted to exhibit minimal mismatching during Golden Gate assembly (GGA), based on the work of Potapov et al. [16]. This set of 4-base junctions included AAGG, ACAA, ACCG, ACTC, AGGA, AGTT, ATGA, CATC, CTCT, CTGG, GACC, GCGA, GGAA, GTAT, TAAT, TACA, TAGA, and TTTG. The complete E2 genome was subdivided into 16 regions of equal size, and the junction points between adjacent regions were noted. The 500 bp regions centered on each of these junction points (−250 bp upstream and +250 bp downstream) were scanned for the presence of one of the 4-base overhangs above. Each junction point was then assigned a single 4-base overhang in the 18-member set above, and this chosen overhang was not re-used for any other junction points. Subgenomic ‘A’ regions were then defined as the nucleotides spanning one 4-base overhang to the next.

The decomposition of each ‘A’ region into ‘B’ subfragments was then carried out with logic analogous to the above, except the regions scanned around each junction point were only 100 bp. Likewise, each junction point was then assigned a single 4-base overhang in the 18-member set above (excepting the two 4-base overhangs already used at the termini of the complete ‘A’ region), and the chosen overhang was not re-used for any other junction points within that particular ‘A’ region. Finally, each ‘B’ subfragment was defined as the nucleotides spanning one 4-base overhang to the next.

### 2.10. Construction of Domesticated Destination Vectors to Enable Golden Gate Assembly

#### 2.10.1. Destination Vectors for ‘A’ Subgenomic Plasmids

The gene fragment pGGBacMed1.Amp, containing a minimized ampicillin resistance and medium-copy pMB1-like origin of replication, was synthesized by Integrated DNA Technologies (Coralville, IA, USA). To obtain a circularized plasmid from this fragment, 20 ng of this synthetic fragment were subjected to ligation with T4 ligase (#M0202T, New England Biolabs, Ipswich, MA, USA) according to the manufacturer’s protocol, followed by transformation into NEB^®^ 5-alpha *E. coli* (#C2987H, New England Biolabs) and the isolation of carbenicillin-resistant colonies. The colonies were amplified, and plasmid was extracted and sequence-confirmed. This synthetic, domesticated medium-copy plasmid was designated pGGBacMed1.Amp. pGGBacMed1.Amp was PCR-amplified using primers LacZcapMedHighgibs_F and LacZcapHighgibs_R and combined with the synthetic gene fragment Pkat_LacZa in Gibson assembly to generate pGGH.LacZα, a domesticated high-copy plasmid encoding a LacZα reporter transcriptionally controlled by an *aphAI* promoter and a his-operon terminator.

For each of the 16 destination vectors to be constructed, a LacZα-encoding PCR product was amplified, flanked by the appropriate 4-base Golden Gate overhangs. Each of the 4-base overhangs (nnnn) was flanked by BsaI sites internally (underlined) and BsmBI sites externally (bolded), as illustrated below.

5′ …**CGTCTCA**nnnnTGAGACC…LacZα cassette…GGTCTCGnnnn**CGAGACG**… 3′

Each PCR product was then combined in Gibson assembly with a second PCR product generated using primers pGG_F and pGGH_R and plasmid pGGBacMed1.Amp as template. Destination vectors were all confirmed through blue colony phenotype and Sanger sequencing. The full set of 4-base overhangs used for each ‘A’ subgenomic region is given in Table 1.

#### 2.10.2. Construction of a Domesticated Bacterial Artificial Chromosome

Domesticated bacterial artificial chromosome (BAC) bGGVoy2 was constructed via the Gibson assembly of two synthetic gene fragments and five PCR products as follows: All fragments used for Gibson assembly lacked the recognition sites of Type IIS REN enzymes BsaI, BsmBI, and PaqCI as well as Type II REN enzymes including, but not limited to, AgeI, KpnI, NheI, and BamHI.

First, an appropriate template for the generation of fragments by PCR was constructed by MluI REN digest of bacmid bMON14272, followed by agarose gel resolution and the extraction of the 13.4 kb band containing the mini-F replicon and kanamycin resistance cassette. The fragment was subjected to a T4 ligase ligation reaction (#M0202T, New England Biolabs) according to the manufacturer’s instructions and transformed into NEB^®^ 10-beta *E. coli* (#C3020K, New England Biolabs). Transformants were plated on LB agar containing kanamycin. A midi-scale culture was amplified from a kanamycin-resistant clone, and the BAC-containing bMluIFrag was successfully extracted.

bMluIFrag was then used as a template in five PCR reactions. PCR products of the expected sizes were resolved via agarose gel electrophoresis, extracted, and combined in Gibson assembly with two synthetic gene fragments to generate the complete BAC. Enzyme sites were removed by the design of the synthetic fragments as well as the Gibson assembly junction points between PCR products. Assembly products were electroporated into NEB^®^ 10-beta *E. coli,* and transformants were plated on LB agar containing kanamycin. BAC DNA was isolated from a kanamycin-resistant clone and designated bGGVoy. To increase the copy number of this domesticated BAC, a dual LacZα reporter and high-copy pUC origin were inserted into bGGVoy. The inserted LacZα and pUC origins were flanked by BsmBI sites to enable excision upon Golden Gate assembly, leaving only the low-copy BAC region harboring the assembled construct. A PCR amplicon, including the LacZα and pUC origins of plasmid, was amplified using the appropriate forward and reverse primers. The PCR product was combined in Gibson assembly with MluI-digested bGGVoy and transformed into NEB^®^ 10-beta *E. coli*. Plasmid DNA was extracted from kanamycin-resistant, X-gal-reactive, LacZα-expressing blue colonies and confirmed via Sanger sequencing. This dual high/low-copy destination vector was designated bGGVoy2.

### 2.11. Golden Gate Assembly-Based Construction of Synthetic Baculovirus Genomes

#### 2.11.1. Chemical Synthesis of DNA and Cloning into Kanamycin-Resistant, BsaI-Lacking Plasmids

BsaI REN recognition site flanks were added to each of the ‘B’ subfragments designed in the preceding section to generate the final sequences to be chemically synthesized. The sequences were submitted to Twist Bioscience (San Francisco, CA, USA) and Genewiz, Inc. (South Plainfield, NJ, USA) for chemical synthesis and cloning. Gene fragments were cloned, respectively, into high-copy plasmids pTwist-Kan and pUC-GW-Kan, encoding kanamycin resistance and lacking BsaI recognition sites. 

#### 2.11.2. Construction of ‘A’ Subgenomic Plasmids by BsaI Golden Gate Assembly

For every ‘A’ region to be constructed, 50 fmol of each ‘B’ subfragment-containing plasmid was combined in a BsaI Golden Gate assembly reaction with 50 fmol of the compatible destination vector in a 30 µL reaction along with 1× NEBridge Ligase Mastermix (#M1100, New England Biolabs) and 1 µL of BsaI-HFv2 enzyme (#R3733, New England Biolabs). The mixture was subjected to thermal cycling as follows: 60 cycles of 37 °C (5 min) at 16 °C (5 min) followed by 1 cycle of 60 °C (5 min) and 10 °C hold. Reaction products were dialyzed against de-ionized H_2_O on 0.025 µm MCE membranes (#VSWP01300 Merck Millipore, Tullagreen, Carringtwohill, Ireland), followed by electroporation into NEB^®^ 10-beta *E. coli*. Transformants were plated on blue/white LB agar plates containing kanamycin, IPTG, and X-gal for selection and blue/white colony screening. Several white colonies were used to inoculate 4 mL LB cultures containing 50 μg/mL kanamycin. 

#### 2.11.3. Construction of Baculovirus Genomes from ‘A’ Subgenomic Plasmids by BsmBI Golden Gate Assembly

50 fmol of each confirmed ‘A’ plasmid was combined in a BsmBI Golden Gate assembly reaction with 50 fmol of destination vector bGGVoy2 in a 30 µL reaction along with 1× NEBridge Ligase Master Mix (#M1100, New England Biolabs) and 1 µL of BsmBI-HFv2 enzyme (#R3733, New England Biolabs). The mixture was subjected to thermal cycling as follows: 60 cycles of 37 °C (5 min) at 16 °C (5 min) followed by 1 cycle of 60 °C (5 min) and 10 °C hold. 

Reaction products were dialyzed against de-ionized H_2_O on 0.025 µm membranes (# Millipore-Sigma), followed by electroporation into CopyCutter^TM^ EPI400^TM^
*E. coli* (#C400EL10, Lucigen Corporation). Transformants were plated on blue/white LB agar plates containing carbenicillin, IPTG, and X-gal for selection and blue/white screening. Miniprep-scale plasmids were extracted from cultures inoculated with white colonies using Monarch^®^ Plasmid Miniprep Kit (#T1010, New England Biolabs) and confirmed by SspI REN digest. 

#### 2.11.4. Incorporation of Two Halves of a Chloramphenicol Resistance Cassette into Two Adjacent ‘A’ Regions to Allow Selection for Full-Length Genome Assembly

In order to counteract the decreased transformation efficiency of large-molecular-weight DNA into *E. coli* [24], we incorporated two halves of a synthetic chloramphenicol resistance cassette that, upon successful assembly, confers chloramphenicol resistance to transformants. We chose the junction point between the A4 and A5 regions, as this junction point lay within the *odv-e66* gene, which is non-essential in *Sf9* insect cell culture [25]. The plasmids containing ‘B’ subfragments B64 and B65 were modified to include the first and second halves of a chloramphenicol resistance cassette, respectively, to obtain modified plasmids B64.3 and B65.3. These two B plasmids, B64.3 and B65.3, were then used in BsaI Golden Gate assembly, as described above, to construct new versions of the A4 and A5 regions, which were used in the construction of both the SynBac1 and SynBac2 designs.

#### 2.11.5. Site-Directed Mutagenesis of ‘B’ Fragments Containing *lef8* and *p49* Sequences

To modify the ‘B’ fragments containing late expression factor 8 *(lef8*) and *p49* gene sequences, a PCR-based site-directed mutagenesis approach was used. PCR reactions were set up using 1 ng of B73 and B230 plasmids as templates, along with primer pairs introducing the desired base modifications. PCR products were subjected to DpnI digest to remove plasmid template, and then subjected to Gibson assembly with the following reaction composition: 1 µL DpnI-digested PCR product, 9 µL H_2_O, and 10 µL 2X HiFi Assembly Master Mix (#E2621, New England Biolabs). The assembly reaction was incubated at 50 °C for 30 min, then transformed into competent NEB^®^ 5-alpha *E. coli* (#C2987H, New England Biolabs) according to the manufacturer’s instructions. Transformants were selected on agar plates containing 150 µg/mL carbenicillin and grown overnight at 37 °C.

Clones were isolated and expanded in 4 mL LB cultures containing 150 µg/mL carbenicillin. Plasmid DNA was extracted from pelleted bacteria using the Monarch^®^ Plasmid Miniprep Kit (#T1010S, New England Biolabs) according to the vendor’s instructions. Plasmids were Sanger-sequence-confirmed using Genewiz universal primer M13-48REV.

#### 2.11.6. Cloning of rAAV Elements into bMON14272 and SynBac2

For rAAV cloning, the bMON14272 bacmid was modified at the chitinase A/viral cathepsin (*chiA/v-cath)* locus to have disrupted *chiA* and *v-cath* genes via homologous recombination in transfected *Sf9* cells with a reporter gene plasmid with unique REN sites flanking the reporter gene. For the RepCap bacmid, a multigene cassette containing AAV1 capsid and AAV2 Rep78 ORFs under regulation of baculovirus p10 and polh promoters, respectively, was inserted into the *chiA/v-cath* locus by REN digest, Gibson assembly and bacmid isolation in bacteria. An independently expressed Rep52 ORF under polh promoter was inserted into the same bacmid in the ecdysosteroid UDP-glycosyltransferase *(egt*) gene locus by digestion with a unique REN, Gibson assembly, and bacmid isolation in bacteria. For the ITR transgene bacmid, a gene cassette containing AVV2 ITR elements surrounding a cytomegalovirus (CMV) promoter and secreted embryonic alkaline phosphatase (SEAP) ORF was cloned into the *chiA/v-cath* locus in the same manner as above.

The SynBac2 virus was engineered to have deleted *chiA/v-cath* genes and a unique REN site in the *egt* gene such that it was equivalent to the designs made to bMON14272 for rAAV cloning. The same RepCap and ITR-SEAP transgene cassettes were inserted directly into the REN-cut SynBac2 bacmid via Gibson Assembly. The Gibson Assembly insertion of Rep78-Cap and Rep52 into their *chiA-vcath* and *egt* locations was performed via the one-step cutting of the bacmid with unique REN enzymes in the *chiA-vcath* and *egt* loci. 

The generation and recovery of Gibson-assembled rAAV RepCap and ITR-SEAP bacmids in *E. coli* took two days. The molar ratios of the rAAV inserts and REN digested bacmids were optimized such that most colonies were positive after Gibson assembly.

## 3. Results

### 3.1. In Silico Domestication and Modification of the AcMNPV Strain E2 Genome

To enable a Golden Gate assembly (GGA)-based approach to synthetic genome construction, we first set out to remove all recognition sites for the Type IIS restriction endonucleases (RENs) BsaI, BsmBI, and PaqCI (Figure 1A). To reduce the risk of disrupting essential *cis* elements, we employed the following rationale. A region comprising 200 bp upstream and downstream of each native REN site was queried against all the Alphabaculovirus genomes in the NCBI database using the blastn algorithm. Each native REN site was then removed by substituting the sequence of a closely related baculovirus genome lacking the REN site, if such a genome was identified (Figure 1B). If several candidate genomes lacking the REN site were identified, generally the genome sharing a higher overall sequence identity compared to the query was chosen as the reference to inform the site removal. Finally, the sequence substitutions conferring no amino acid changes to the known open reading frames were prioritized over those conferring amino acid changes. To enable further differentiation between natural baculovirus genomes and this synthetic genome, the same rationale above was used to remove the recognition sites for an additional panel of Type II restriction endonucleases. As a final modification, we removed the *chiA/v-cath* locus, as these genes are known to be dispensable for baculovirus replication and are otherwise undesirable for the production of biologics [26]. 

### 3.2. Division of the Domesticated AcMNPV Genome Design into Subgenomic ‘A’ Regions and ‘B’ Subfragments

Golden Gate assembly (GGA) is routinely used to achieve a high-fidelity, ordered assembly of 5–10 DNA parts, and recent optimizations of the GGA design and reagents have enabled the assembly of as many as 52 DNA parts [16,17,18]. In principle, therefore, our domesticated genome could be constructed in a single-step assembly of 30–50 DNA parts, each roughly 3–5 kb. However, we instead chose an assembly approach comprising two hierarchical steps of assembly. This hierarchical assembly reduced the size of the DNA fragments needed to be chemically synthesized and translated to shorter synthesis times, a lower synthesis cost, and a higher fidelity of the synthesis. We first subdivided the domesticated genome into 16 subgenomic regions (denoted ‘A’ regions), each comprising 8–9 kb of the sequence. Then, each of these ‘A’ regions were further subdivided into 16 subfragments (denoted ‘B’ subfragments’), each comprising 300–750 bp of the sequence. Thus, our assembly strategy demanded a total of 16 × 16 = 256 DNA fragments to be chemically synthesized. Adjacent ‘A’ subgenomic regions were overlapped at their ends by unique 4 bp sequences that would ultimately serve as the 5′ overhangs to direct ordered fragment annealing during BsmBI-based GGA (see Table 1). Likewise, adjacent ‘B’ subfragments within each ‘A’ fragment were also overlapped at their ends by unique 4 bp sequences that would guide ordered annealing during BsaI-based GGA. An illustration of the 4 bp overlaps between adjacent ‘A’ regions and adjacent ‘B’ subfragments is given in Figure 2. The overall two-step hierarchical assembly design is illustrated in Figure 3. 

### 3.3. Construction and Prototyping of a Domesticated AcMNPV Genome Derived Entirely from Chemically Synthesized DNA Sequences

All 256 ‘B’ subfragments that were flanked by appropriately oriented BsaI recognition sites were chemically synthesized and cloned into Golden Gate assembly (GGA)-compatible plasmid vectors encoding kanamycin resistance. The 16 ‘A’ regions, denoted A1 through A16, were constructed via BsaI GGA and ultimately captured in custom ampicillin-resistant destination plasmid vectors, comprising appropriately positioned BsmBI recognition sites flanking each ‘A’ region. Finally, the plasmids harboring regions A1 through A16 were combined in a BsmBI GGA to generate the synthetic *Ac*MNPV genome, which was captured in a GGA-compatible bacterial artificial chromosome (BAC). The sequence integrity of the synthetic *Ac*MNPV genome, which we denoted SynBac1, was confirmed via Illumina sequencing. We further exploited the extensive removal of REN recognition sites to readily discriminate SynBac1 from bMON14272 [5], a BAC harboring the non-synthetic genome of the *Ac*MNPV strain E2 (Figure 4). While SynBac1 and bMON14272 exhibited similar DNA fragment banding patterns in agarose gel electrophoresis upon HindIII and MfeI digestion, SynBac1 was completely resistant to cleavage by six illustrative restriction endonucleases—in total demonstrating the removal of 99 recognition sites found in bMON14272.

Upon the transfection of the *Sf9* insect cells with SynBac1 DNA, we observed a detectable baculovirus infectious titer in the culture supernatant. However, a comparison of replication kinetics between SynBac1 and bMON14272 revealed a 10-fold replication defect for SynBac1 (Figure 5), suggesting deleterious variants or elements present in the SynBac1 genome. 

### 3.4. Identification of Potentially Deleterious Sequence Variations in Essential Genes lef8 and p49 Inherited by SynBac1

To ensure that there were no unexpected sequence variations between bMON14272 and SynBac1, we subjected bacmid bMON14272 to Illumina sequencing and aligned reads to the *Ac*MNPV strain E2 genome (Accession KM667940.1) (Figure 6). Surprisingly, we observed two non-silent variations in the *lef8* and *p49* genes, resulting in a lysine and proline deletion, LEF8 and P49 proteins respectively, relative to the NCBI record. We likewise observed these two discrepancies when subjecting DNA extract from wild-type *Ac*MNPV E2-infected *Trichoplusia ni* cells (ATCC # VR-1344) to Illumina sequencing. Notably, the *lef8* and *p49* sequences we observed in our analysis of bMON14272 and wild-type *Ac*MNPV E2 matches the *lef8* and *p49* sequences in the NCBI record for the C6 strain (Accession NC_001623.1) [27], whereas the extra LEF8 lysine and extra P49 proline in the NCBI record KM667940.1 do not appear in any homologs of closely related baculoviruses. Importantly, LEF8 and P49 proteins are known to fulfill essential baculovirus functions [28,29,30,31]. We therefore hypothesized that the replication defect in SynBac1 could have been caused by the use of the *lef8* and *p49* sequences represented in the NCBI record KM667940.1, which may not be the correct sequences of these genes in the wild-type *Ac*MNPV strain E2. 

### 3.5. Synthetic SynBac2 Genome Exhibits Comparable Replication Kinetics and Recombinant AAV Productivity to Non-Synthetic Baculovirus Genomes

We modified the design of the *lef8* and *p49* loci of SynBac1 to instead reflect the amino acid sequence of these genes in the C6 strain. We designated this modified genome design SynBac2. The ‘B’ plasmids corresponding to the problematic regions of *lef8* and *p49* were corrected via PCR-based site-directed mutagenesis and incorporated into updated ‘A’ regions, which were combined with the unmodified ‘A’ regions in the BsmBI GGA to yield the SynBac2 BAC. We then compared the replication kinetics of bMON14272, SynBac1, and SynBac2 after the BAC DNA transfection into *Sf9* insect cell culture as well as after high-MOI infection (Figure 7). Whereas SynBac1 exhibited defective replication kinetics relative to bMON14272, SynBac2 replicated indistinguishably from bMON14272. These results demonstrate the successful construction of a baculovirus genome derived entirely from chemically synthesized DNA that exhibits normal replication kinetics in insect cell cultures. Secondly, these results strongly suggest that the replication defect in the prototype SynBac1 design is due to amino acid variations in LEF8, P49, or both.

Having confirmed the integrity of SynBac2 for baculovirus replication, we next investigated whether SynBac2 exhibited a similar capability for biologic production compared to a non-synthetic genome. We inserted adeno-associated virus (AAV) *rep* and *cap* expression elements into SynBac1, SynBac2, and a non-synthetic control bacmid. Likewise, we inserted a SEAP reporter transgene flanked by AAV ITRs into each of these bacmid designs (Figure 8A). Upon the co-infection of the *Sf9* insect cell cultures with RepCap- and ITR-baculoviruses, we observed that SynBac2 supported rAAV-SEAP production comparably to the non-synthetic bacmid design (Figure 8B). A Western blot analysis further confirmed the presence of AAV Rep and Cap proteins during the production process (Figure 8C).

## 4. Discussion

In this study, we used Golden Gate assembly (GGA) to successfully construct a large (>130 kb) virus genome derived entirely from non-templated, chemically synthesized DNA fragments. One of the major challenges in assembling synthetic virus genomes is the high degree of sequence complexity and variability. Baculoviruses are known for their AT-rich genomes, comprising many repetitive sequences, which can lead to errors in DNA synthesis and fragment assembly. To mitigate these challenges, we chose to minimize the size of the individual DNA fragments (300–750 bp) to decrease the DNA synthesis turn-around time and failure rate.

GGA has the advantage of assembling up to 52 fragments [16,17,18], although the fidelity and efficiency of the assembly are generally higher at lower fragment counts. GGA generally requires the absence of recognition sites for the Type IIS REN used for every assembly step. Therefore, it was important that we remove these REN sites while minimizing the risk of negatively impacting baculovirus replication or gene expression.

We employed a BLAST-based strategy to mitigate the risk of mutating a series of REN recognition sites across the genome. The removal of these restriction enzyme sites enabled the use of a two-step hierarchical assembly, the first step based on BsaI GGA and the second step based on BsmBI GGA. Furthermore, the removal of these and other REN sites allowed us to physically distinguish between our synthetic genome and the derivatives from any functionally similar but non-synthetic genomes.

The two-step hierarchical assembly of 16 fragments at each step enabled the use of the smaller DNA parts that are more amenable to chemical synthesis (16 × 16 = 256 total fragments). Noting the success of Pryor and colleagues [18] in constructing the 40 kb genome of phage T7 from 52 fragments, we envision that future baculovirus genome assembly schema may employ a single-step rather than two-step hierarchical assembly—though this would necessitate the use of larger fragments, e.g., 2–5 kb, which may be more time-consuming to synthesize. The highly complex assembly performed by the authors above necessitated the use of a stringent selection step—the reconstitution of infectious phage—to isolate the correctly assembled genomes. In the case of the baculovirus, the direct transfection of the assembly products into cultured insect cells could serve as an analogous stringent selection step.

Upon the assembly and sequence-confirmation of our first complete baculovirus genome design, SynBac1, we found that infectious baculovirus particles were produced after the transfection of *Sf9* insect cells with SynBac1 DNA extracted from *E. coli*. However, we also observed that SynBac1 yielded a 10-fold less-infectious titer compared to the non-synthetic control, bacmid bMON14272 (Figure 5). While we initially presumed the replication defect in SynBac1 was due to one of the silent mutations in our initial genome design, an independent re-analysis of the *Ac*MNPV E2 sequence revealed another possibility. After sequencing the wild-type *Ac*MNPV E2 DNA, as well as BAC bMON1472, which comprises the *Ac*MNPV E2 sequence 5, and aligning the reads to NCBI record KM667940.1 [19], we found that KM667940.1 potentially contained errors in the essential genes *lef8* and *p49* (Figure 6). SynBac1, whose design is ultimately based on KM667940.1, therefore would have inherited these likely erroneous *lef8* and *p49* sequences. In the second genome design, SynBac2, we modified the *lef8* and *p49* genes to instead reflect the sequence attested by our *Ac*MNPV E2 sequence re-analysis, and this modification resulted in replication kinetics indistinguishable from those of bMON14272. To our knowledge, this represents the first generation of an infectious baculovirus from a genome entirely derived from non-templated, chemically synthesized DNA.

As well as demonstrating the feasibility of assembling a large synthetic DNA virus with hundreds of small fragments via Golden Gate assembly, we also showed one of the major drawbacks, that being the hundreds of silent mutations that needed to be engineered into the genome to remove type II and type IIS REN sites. When SynBac1 was completed, we were unsure as to whether one or more of the silent mutations had made it deficient. We could not observe SynBac1′s deficiency until the very end of its in vitro construction and transfection into insect cells. The disruption of the cryptic promoter elements could have been the cause. We were fortunate that baculoviruses are well characterized and that we had a pool of genomic data to draw upon to identify the *lef8* and *p49* anomalies. A lesser-characterized DNA virus type may not be an ideal candidate for this kind of synthetic biology.

Finally, to verify the integrity of very-late gene expression and baculovirus helper functions, we set out to produce a recombinant adeno-associated virus (rAAV) with our synthetic genome designs. We inserted AAV Rep and Cap expression cassettes, as well as transgenes flanked by AAV ITRs, into SynBac1, SynBac2, and the non-synthetic control bacmid. Upon the co-infection of the *Sf9* insect cells with RepCap and ITR baculoviruses, we observed that SynBac2 exhibited rAAV production kinetics comparable to the non-synthetic control bacmid.

Overall, the successful assembly of a synthetic virus genome represents a major milestone in the field of synthetic biology, with significant implications for our understanding of virus replication and pathogenesis. This work also highlights the potential for synthetic biology to contribute to the development of novel therapeutics and vaccines for a range of viral diseases.

## Figures and Tables

**Figure 1 viruses-15-01981-f001:**
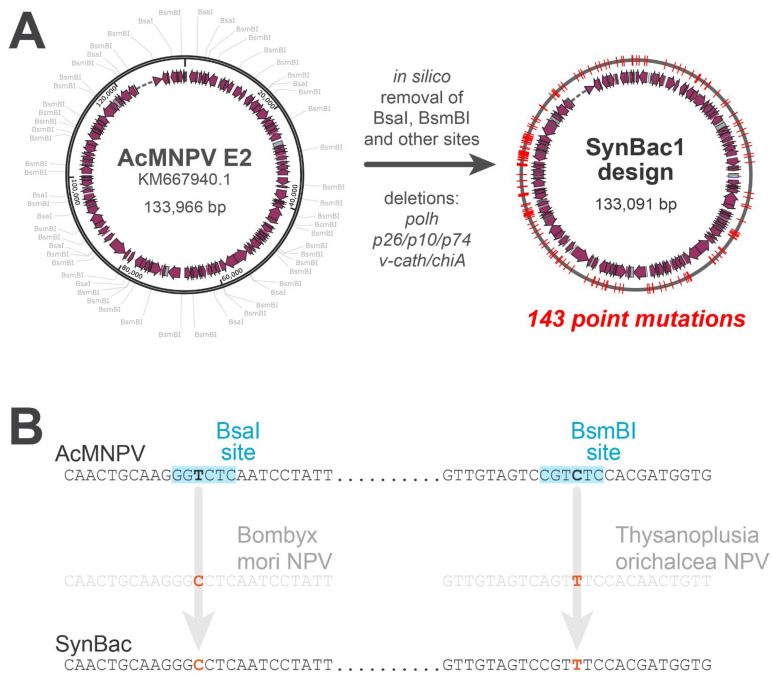
In silico design and domestication of a synthetic baculovirus genome. (**A**) Drafting of a modified *Ac*MNPV genome compatible with Golden Gate assembly (GGA). Recognition sites for Type IIS enzymes BsaI, BsmBI, and PaqCI were first removed in silico to enable construction of the genome by GGA. Recognition sites of the following restriction enzymes were also removed to enable future cloning strategies as well as to differentiate SynBac1 and SynBac2 from non-synthetic genomes: AbsI, AgeI, ApaI, AscI, AsiSI, AvrII, BamHI, BbvCI, Eco53kI, EcoNI, FseI, KpnI, NheI, NotI, PmeI, SmaI, and StuI. (**B**) To mitigate the risk of mutating cis-elements critical for baculovirus function, whenever possible, each enzyme site was mutated by cross-referencing closely related baculovirus genomes that lacked the enzyme site. Silent mutations within known open reading frames were preferred over non-silent mutations.

**Figure 2 viruses-15-01981-f002:**
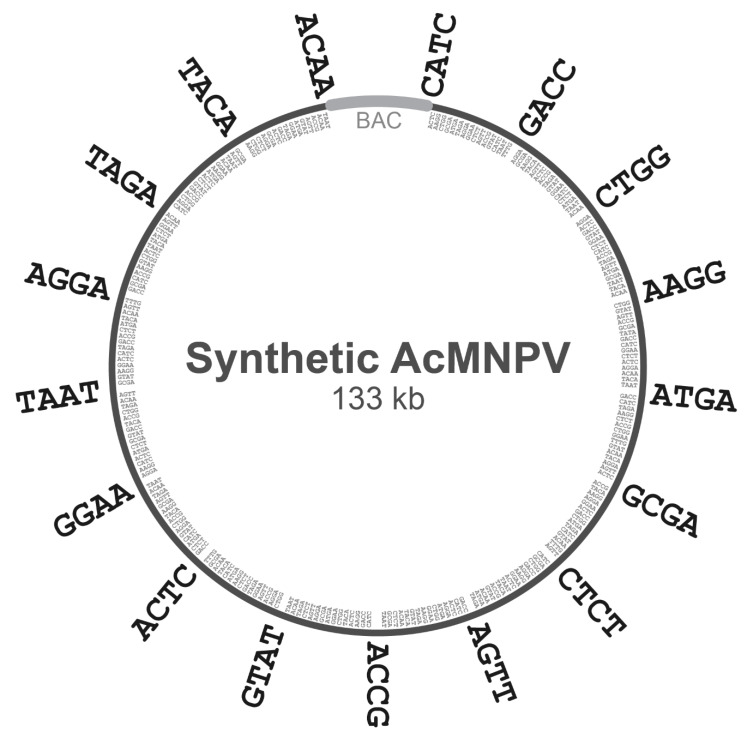
Division of domesticated genome into subgenomic ‘A’ regions and ‘B’ subfragments. In preparation for Golden Gate assembly (GGA)-based construction of a synthetic *Ac*MNPV genome, the domesticated *Ac*MNPV genome of Figure 1 was then subdivided into 16 ‘A’ regions, roughly 8–9 kb in size, each terminated by the 4-base junctions indicated on the outside of the circular genome depicted above. The 4-base junctions between ‘A’ regions were chosen to maximize ligation fidelity during GGA. Each of these ‘A’ regions was then further subdivided into 16 ‘B’ subfragments, roughly 300–750 bp in size. Each ‘B’ subfragment was likewise terminated by 4-base junctions, indicated on the inside of the circular genome above. The 4-base junctions between each set of 16 ‘B’ subfragments were chosen to maximize ligation fidelity during GGA.

**Figure 3 viruses-15-01981-f003:**
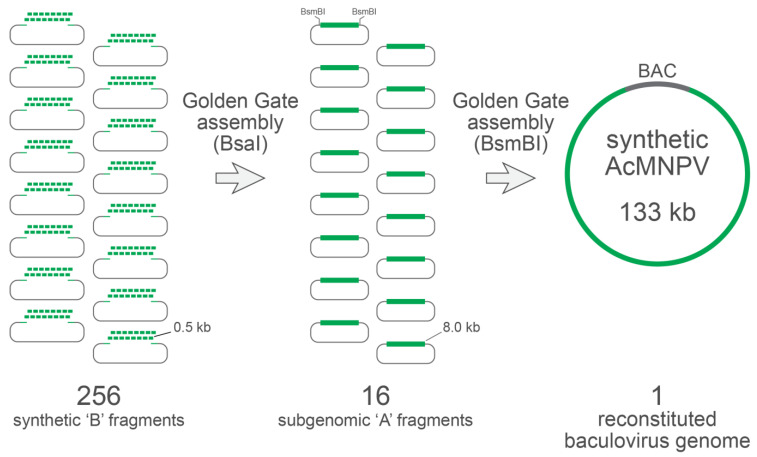
Schema for two-step assembly of modified *Ac*MNPV genome entirely from chemically synthesized DNA fragments. All ‘B’ subfragments (256 total, roughly 0.5 kb each) were chemically synthesized, cloned into plasmids encoding kanamycin resistance, and sequence confirmed. Each ‘A’ region was constructed by the BsaI-based Golden Gate assembly (GGA) of each set of 16 ‘B’ fragments into a destination plasmid encoding ampicillin resistance and including BsmBI sites flanking the assembled ‘A’ insert (roughly 8 kb). Finally, the complete synthetic *Ac*MNPV genome was assembled via the BsmBI-based GGA of all 16 ‘A’ regions into a bacterial artificial chromosome lacking BsmBI sites and encoding kanamycin resistance.

**Figure 4 viruses-15-01981-f004:**
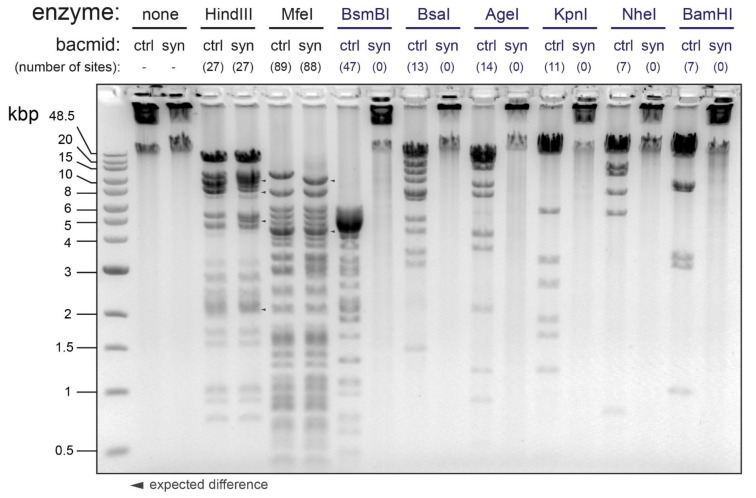
Restriction enzyme-based demonstration that the SynBac1 genome is derived entirely from chemically synthesized sequences. Bacmids bMON14272 (ctrl) and SynBac1 (syn) were subjected to digestion by a panel of restriction enzymes. HindIII and MfeI were used to confirm the overall integrity of the SynBac1 genome by similar banding pattern to that of bMON14272. Expected banding differences due to deletion of *chiA*/*v-cath* were indicated by triangles. BsmBI, BsaI, AgeI, KpnI, NheI, and BamHI were used to demonstrate the successful removal of 99 enzyme sites throughout the SynBac1 genome.

**Figure 5 viruses-15-01981-f005:**
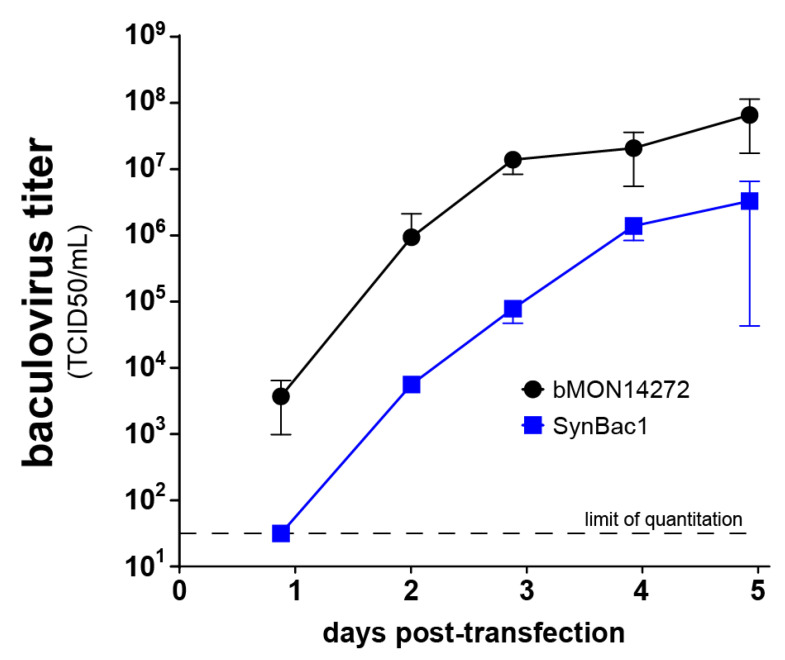
SynBac1 genome is replication-competent but exhibits a 10-fold replication defect relative to bMON14272. bMON14272 and SynBac1 bacmid DNA was isolated from two independent *E. coli* clones. Each preparation of bacmid DNA was transfected into *Sf9* insect cell culture, and the kinetics of the budded virus output were quantitated via TCID50 assay. Error bars depict standard error of the mean between duplicate transfected cultures.

**Figure 6 viruses-15-01981-f006:**
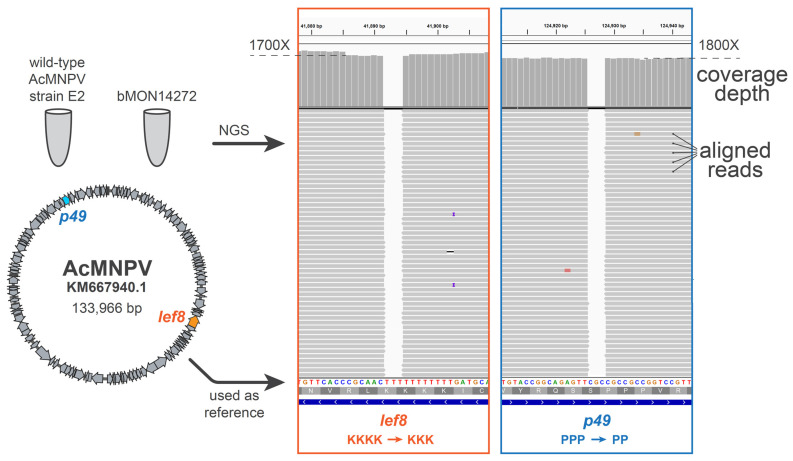
Errors or variations observed in the *lef8* and *p49* genes of *Ac*MNPV (strain E2) NCBI record KM667940.1. Whole DNA extract was prepared from *Ac*MNPV strain E2-infected *Trichoplusia ni* cells (#VR-1344, ATCC). DNA extract, as well as E2-derived bacmid bMON14272, were subjected to Illumina whole genome sequencing (SeqCenter, LLC, Pittsburgh, PA, USA), and reads were aligned to record KM667940.1. Aligned reads within essential genes *lef8* and *p49* were visualized using Integrated Genome Viewer, and non-silent variations from the reference were noted. Depicted reads are specifically from E2-infected *Trichoplusia ni* cells, with view zoomed on KM667940.1 nucleotide positions 41,879 to 41,908 (*lef8*) and nucleotide positions 124,912 to 124,942 (*p49*).

**Figure 7 viruses-15-01981-f007:**
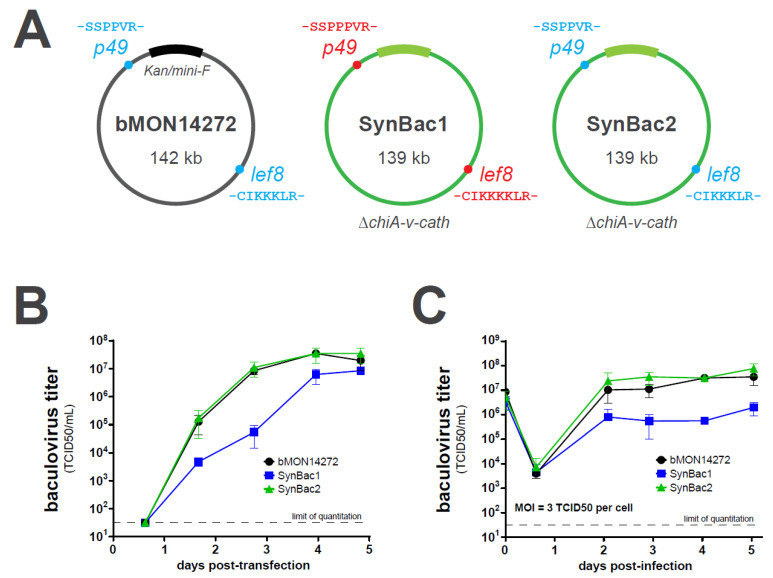
SynBac2 genome replicates in *Sf9* cells comparably to bMON14272. (**A**) The respective B subfragments encoding the lysine quadruplet in LEF8 and the proline triplet in P49 were modified to match the amino acid sequence of these genes in bMON14272 and our sequence-confirmed wild-type *Ac*MNPV E2 sample. These new B subfragments were used to assemble new A regions and a second synthetic baculovirus genome known as SynBac2. (**B**) Bacmid preps isolated from three independent *E. coli* clones were transfected into *Sf9* cells, and the resulting budded virus was quantitated via TCID50 assay. Error bars indicate standard deviation between cultures transfected by the three bacmid preps for each design. (**C**) To obtain single-cycle replication kinetics, the budded virus-containing supernatant from (**B**) was used to infect new *Sf9* cultures at a multiplicity of infection of 3 TCID50 per cell. The resulting budded virus output was quantitated via TCID50 assay. Error bars indicate standard deviation between cultures infected by the triplicate baculovirus stocks obtained from the three independent bacmid preps.

**Figure 8 viruses-15-01981-f008:**
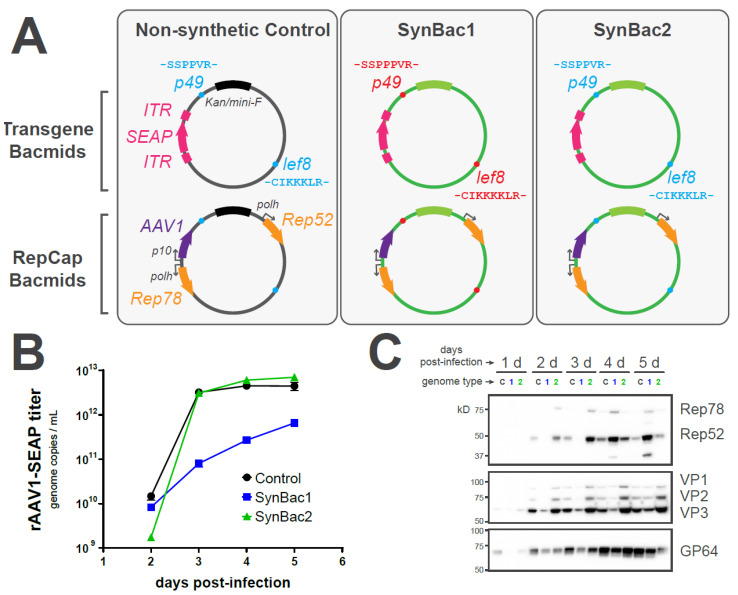
SynBac2 genome supports rAAV production comparatively to a non-synthetic bacmid. (**A**) Illustration of bacmid designs in this experiment. Amino acid contexts of the LEF8 lysine tract and the p49 proline tract are indicated for each bacmid. rAAV elements, including ITR-flanked SEAP reporter transgene as well as expression cassettes for AAV1 Cap and AAV2 Rep78/Rep52. All bacmids included in the deletions of the *chiA*-*v-cath* and *p26*/*p10*/*p74* gene clusters were inserted either at the *chiA/v-cath* or *egt* locus. (**B**) Quantitation of packaged rAAV1-SEAP vector genomes in *Sf9* lysate after infection by synthetic and non-synthetic bacmid-derived virus. *Sf9* cultures were infected at MOI = 0.002 with dual transgene and RepCap baculoviruses of the indicated type. At the indicated time-points, cells were detergent-lysed, and DNase-resistant vector genome copies were analyzed via qPCR. (**C**) Western blot analysis of AAV Rep, AAV Cap, and *Ac*MNPV GP64 levels at the indicated times post-infection. Baculovirus genome type for each lane is indicated by “C” (non-synthetic control), “1” (SynBac1), or “2” (SynBac2).

**Table 1 viruses-15-01981-t001:** Left and right 4-base ends used for Golden Gate assembly of each subgenomic ‘A’ region.

Subgenomic ‘A’ Region	Left 4-Base Motif (5′ to 3′)	Right 4-Base Motif (5′ to 3′)
A1	ACAA	GACC
A2	GACC	CTGG
A3	CTGG	AAGG
A4	AAGG	ATGA
A5	ATGA	GCGA
A6	GCGA	CTCT
A7	CTCT	AGTT
A8	AGTT	ACCG
A9	ACCG	GTAT
A10	GTAT	ACTC
A11	ACTC	GGAA
A12	GGAA	TAAT
A13	TAAT	AGGA
A14	AGGA	TAGA
A15	TAGA	TACA
A16	TACA	CATC

## Data Availability

Additional data require confidential disclosure agreement (CDA) with Voyager Therapeutics.

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
