# Peer review of "A Baculovirus Expression Vector Derived Entirely from Non-Templated, Chemically Synthesized DNA Parts"

_viruses, 2023, doi:10.3390/v15101981_

Round 1

Reviewer 1 Report

The manuscript describes the construction of the complete genome of the prototype baculovirus Autographa californica multiple nucleopolyhedrovirus (AcMNPV) based totally on synthetic DNA. This was accomplished by assembling multiple DNA fragments using Golden Gate Cloning. The artificial virus was designated SynBac and compared to a standard baculovirus expression vector resulting in no apparent difference in virus growth and recombinant protein production.

The authors describe a new means of manipulating baculovirus genomes using an in vitro assembly system. The work is certainly of interest for researchers in the field. It is the first example of a fully synthetic baculovirus vector for the generation of theoretically unlimited configurations. I am wondering what is the typical timeline for the construction of a new virus based on the SynBac backbone as compared to flashBac or Bac-to-Bac?

Since the whole process is completed in vitro, non-functional virus can only be identified after transfection into insect cells. I suggest to discuss also drawbacks of the technology in the manuscript.

Minor issues:

nomenclature: SynBac and SynBac1 are used for the same Bacmid vector. That needs to be standardized

line 476: ‘oursequence-confirmed’  change to  our sequence-confirmed’

lines 488-491: insertion of AAV elements and SEAP reporter gene are depicted in Fig. 8A. The details for the construction are not described in material and methods and should be added.

line 501: delete  were inserted either at the v-cath or egt locus.’

The discussion partly rephrases the results and could be condensed (e.g. section errors in lef8 and p49)

References:

Several baculovirus cloning methods are mentioned in the introduction. As for “direct cloning and ligation” the first example was published two years earlier.

Ernst, W., Grabherr, R., Katinger, H. Direct cloning into the Autographa californica nuclear polyhedrosis virus for generation of recombinant baculoviruses. Nucleic Acids Res. 14, 2855-2856 (1994).

Author Response

The manuscript describes the construction of the complete genome of the prototype baculovirus Autographa californica multiple nucleopolyhedrovirus (AcMNPV) based totally on synthetic DNA. This was accomplished by assembling multiple DNA fragments using Golden Gate Cloning. The artificial virus was designated SynBac and compared to a standard baculovirus expression vector resulting in no apparent difference in virus growth and recombinant protein production.

 The authors describe a new means of manipulating baculovirus genomes using an in vitro assembly system. The work is certainly of interest for researchers in the field. It is the first example of a fully synthetic baculovirus vector for the generation of theoretically unlimited configurations. I am wondering what is the typical timeline for the construction of a new virus based on the SynBac backbone as compared to flashBac or Bac-to-Bac?

SynBac A and B fragment assembly is the method of making the backbone of the baculovirus cloning system. A and B fragment assembly is not used for making recombinant baculoviruses with foreign gene inserts like flashBac or Bac-to-Bac backbones. We engineered both SynBac2 and bMON14272 (equivalent to Bac-to-Bac) to have unique REN sites that allowed us to use Gibson assembly to insert foreign genes into these bacmids. Gibson assembly and recombinant AAV bacmid generation in E.coli took about two days. We found Gibson assembly to be fast and reliable and abandoned the use of Tn7 transposition for Bac-to-Bac or homologous recombination Sf9 cells in flashBac years ago.

I have added the following section to the materials and methods. 

2.11.6 Cloning of rAAV Elements into bMON14272 and SynBac2

For rAAV cloning, the bMON14272 bacmid was modified at the chiA-v-cath locus to have disrupted chiA-v-cath genes by homologous recombination in transfected Sf9 cells with a reporter gene plasmid with unique REN sites flanking the reporter gene. For the RepCap bacmid, a multigene cassette containing AAV1 capsid and AAV2 Rep78 ORFs under regulation of baculovirus p10 and polh promoters respectively was inserted into the chiA-v-cath locus by REN digest, Gibson assembly and bacmid isolation in bacteria. An independently expressed Rep52 ORF under polh promoter was inserted into the same bacmid in the egt gene locus by digestion with a unique REN, Gibson assembly and bacmid isolation in bacteria. For the ITR transgene bacmid, a gene cassette containing AVV2 ITR elements surrounding a cytomegalovirus (CMV) promoter and secreted embryonic alkaline phosphatase (SEAP) ORF was cloned into the chiA-v-cath locus in the same manner as above.

The SynBac2 virus was engineered to have deleted chiA-v-cath genes and a unique REN site in the egt gene such that it was equivalent to the designs made to bMON14272 for rAAV cloning. The same RepCap and ITR-SEAP transgene cassettes were inserted directly into the REN cut SynBac2 bacmid by Gibson Assembly. Gibson Assembly insertion of Rep78-Cap and Rep52 into their chiA-vcath and egt locations was done in one step cutting the bacmid with unique REN enzymes in chiA-vcath and egt loci.

The generation and recovery of Gibson assembled rAAV RepCap and ITR-SEAP bacmids in E.coli took two days. Molar ratios of rAAV inserts and REN digested bacmids were optimized such that most colonies were positive after Gibson assembly.

Since the whole process is completed in vitro, non-functional virus can only be identified after transfection into insect cells. I suggest to discuss also drawbacks of the technology in the manuscript.

Added to the Discussion

As well as demonstrating the feasibility of assembling a large synthetic DNA virus with hundreds of small fragments by Golden Gate assembly we also show one of the major drawbacks. That being the hundreds of silent mutations that needed to be engineered into the genome to remove type II and type IIS REN sites. When SynBac1 was completed, we were unsure as to whether one or more of the silent mutations had made it deficient. We could not observe SynBac1’s deficiency until the very end of its in vitro construction and transfection into insect cells. Disruption of cryptic promoter elements could have been the cause. We were fortunate that baculoviruses are well characterized and that we had a pool of genomic data to draw upon to identify lef8 and p49 anomalies. A lesser characterized DNA virus type may not be an ideal candidate for this kind of synthetic biology.

Minor issues:

 nomenclature: SynBac and SynBac1 are used for the same Bacmid vector. That needs to be standardized

Figure 1 legend

 cloning strategies as well as differentiate SynBac from non-synthetic genomes

 cloning strategies as well as differentiate SynBac1 and SynBac2 from non-synthetic genomes

Figure 4 legend

demonstration that SynBac genome is derived

demonstration that the SynBac1 genome is derived

Bacmids bMON14272 (ctrl) and SynBac (syn) were

Bacmids bMON14272 (ctrl) and SynBac1 (syn) were

of the SynBac genome by similar banding

of the SynBac1 genome by similar banding

throughout the SynBac genome.

throughout the SynBac1 genome.

line 476: ‘oursequence-confirmed’  change to  ‘our sequence-confirmed’

Done 

lines 488-491: insertion of AAV elements and SEAP reporter gene are depicted in Fig. 8A. The details for the construction are not described in material and methods and should be added.

Materials & Methods added the following

I have added the following section to the materials and methods. 

2.11.6 Cloning of rAAV Elements into bMON14272 and SynBac2

Shown above

 line 501: delete  ‘were inserted either at the v-cath or egt locus.’

Done 

 The discussion partly rephrases the results and could be condensed (e.g. section errors in lef8 and p49)

The authors wish to retain this rephrasing because it emphasizes a major pitfall in synthetic biology of relying solely on a single published sequence as a template. Elucidation of the mutations in lef8 and p49 took months of work and nearly ended this endeavor were it not for persistence of the first author.

References:

 Several baculovirus cloning methods are mentioned in the introduction. As for “direct cloning and ligation” the first example was published two years earlier.

 Ernst, W., Grabherr, R., Katinger, H. Direct cloning into the Autographa californica nuclear polyhedrosis virus for generation of recombinant baculoviruses. Nucleic Acids Res. 14, 2855-2856 (1994).

I deleted the Lu & Miller reference an

Reviewer 2 Report

The manuscript describes exceptional work in which a baculovirus bacmid is constructed from chemical synthetic sequences. Many restriction sites as well as some non-essential regions in the genome were removed. Mutations in lef8 and p49 were corrected and the corrected synthetic virus produced rAAV particles with identical efficiency as non-synthetic bacmid.

Of interest would be the efficiency of the procedure. After the final Gibson assembly of the complete synthetic bacmid, how many colonies were obtained by transformation and what was the proportion of assemblies that were entirely correct? How was the screening performed? Similarly, how efficient was the assembly of the intermediary B and A fragments.

Details are missing for the construction of the recombinant synthetic bacmid that expresses rAAV. Was this bacmid also constructed de novo from B and A fragments? Could the efficiency of the procedure be improved by starting with a large backbone synthetic fragment into which a rather small fragment with the expression cassette can be assembled?

Author Response

The manuscript describes exceptional work in which a baculovirus bacmid is constructed from chemical synthetic sequences. Many restriction sites as well as some non-essential regions in the genome were removed. Mutations in lef8 and p49 were corrected and the corrected synthetic virus produced rAAV particles with identical efficiency as non-synthetic bacmid.

Of interest would be the efficiency of the procedure. After the final Gibson assembly of the complete synthetic bacmid, how many colonies were obtained by transformation and what was the proportion of assemblies that were entirely correct? How was the screening performed? Similarly, how efficient was the assembly of the intermediary B and A fragments.

B and A fragments were assembled by Gold Gate assembly which is very different from Gibson assembly. With Gold Gate assembly, the type II and type IIS enzymes in the assembly reaction mixture eliminate unassembled constructs. The clone efficiency was near 100%. In instances where we were not getting 100%, we simply supplemented with the type II and type IIS enzymes to ensure 100% recovery of assembled fragments.

Gibson Assembly was used to make some of the plasmid backbones and to clone in rAAV elements into completed SynBac2 and bMON14272 bacmid backbones. I’ve added a description in materials methods. Success in Gibson Assembly requires optimization of REN cut bacmid and gene insert ratios. Efficiency of positive clones was dependent on that.  

Details are missing for the construction of the recombinant synthetic bacmid that expresses rAAV. Was this bacmid also constructed de novo from B and A fragments? Could the efficiency of the procedure be improved by starting with a large backbone synthetic fragment into which a rather small fragment with the expression cassette can be assembled?

SynBac A and B fragment assembly is the method of making the backbone of the baculovirus cloning system. A and B fragment assembly is not used for making recombinant baculoviruses with foreign gene inserts like flashBac or Bac-to-Bac backbones. We engineered both SynBac2 and bMON14272 (equivalent to Bac-to-Bac) to have unique REN sites that allowed us to use Gibson assembly to insert foreign genes into these bacmids. Gibson assembly and recombinant AAV bacmid generation in E.coli took about two days. We found Gibson assembly to be fast and reliable and abandoned the use of Tn7 transposition for Bac-to-Bac or homologous recombination Sf9 cells in flashBac years ago.

I added the following to materials and methods to clarify.

2.11.6 Cloning of rAAV Elements into bMON14272 and SynBac2

For rAAV cloning, the bMON14272 bacmid was modified at the chiA-v-cath locus to have disrupted chiA-v-cath genes by homologous recombination in transfected Sf9 cells with a reporter gene plasmid with unique REN sites flanking the reporter gene. For the RepCap bacmid, a multigene cassette containing AAV1 capsid and AAV2 Rep78 ORFs under regulation of baculovirus p10 and polh promoters respectively was inserted into the chiA-v-cath locus by REN digest, Gibson assembly and bacmid isolation in bacteria. An independently expressed Rep52 ORF under polh promoter was inserted into the same bacmid in the egt gene locus by digestion with a unique REN, Gibson assembly and bacmid isolation in bacteria. For the ITR transgene bacmid, a gene cassette containing AVV2 ITR elements surrounding a cytomegalovirus (CMV) promoter and secreted em-bryonic alkaline phosphatase (SEAP) ORF was cloned into the chiA-v-cath locus in the same manner as above.

The SynBac2 virus was engineered to have deleted chiA-v-cath genes and a unique REN site in the egt gene such that it was equivalent to the designs made to bMON14272 for rAAV cloning. The same RepCap and ITR-SEAP transgene cassettes were inserted di-rectly into the REN cut SynBac2 bacmid by Gibson Assembly. Gibson Assembly insertion of Rep78-Cap and Rep52 into their chiA-vcath and egt locations was done in one step cutting the bacmid with unique REN enzymes in chiA-vcath and egt loci.

The generation and recovery of Gibson assembled rAAV RepCap and ITR-SEAP bacmids in E.coli took two days. Molar ratios of rAAV inserts and REN digested bacmids were optimized such that most colonies were positive after Gibson assembly.